# Outpatient Satisfaction with Tertiary Hospitals in China: The Role of Sociodemographic Characteristics

**DOI:** 10.3390/ijerph16193518

**Published:** 2019-09-20

**Authors:** Linlin Hu, Bright P. Zhou, Shiyang Liu, Zijuan Wang, Yuanli Liu

**Affiliations:** 1School of Public Health, Chinese Academy of Medical Sciences & Peking Union Medical College, 5 Dongdansantiao, Dongcheng District, Beijing 100730, China; hulinlin@sph.pumc.edu.cn (L.H.); lsy@student.pumc.edu.cn (S.L.); wzj1028@student.pumc.edu.cn (Z.W.); 2Stanford University School of Medicine, 291 Campus Drive, Stanford, CA 94305, USA; brightz@stanford.edu

**Keywords:** Chinese healthcare, outpatient, patient satisfaction, tertiary hospital, sociodemographic characteristics, multilevel logistic regression

## Abstract

China’s increasing attention to patient satisfaction evaluation is part of an international trend of patient-centered healthcare. Patient sociodemographic characteristics are important intrinsic factors that will influence satisfaction. This paper aims to better understand how sociodemographic factors affect Chinese patient satisfaction with tertiary outpatient services using data from the 2017 China National Patient Survey. A total of 28,760 outpatient survey responses were analyzed, spanning 136 tertiary hospitals across 31 provinces. Multilevel logistic regression with fixed hospital effects was used to examine the association of patient satisfaction across multiple healthcare domains with sociodemographic factors. Results show that patients who were of a migrant population, of highest income, most educated, and who had medical aid insurance reported the lowest levels of overall satisfaction. Specifically, increasing age was correlated with decreased satisfaction in process management and affordability domains, while high-income and high-education outpatients reported lower satisfaction scores in the hospital environment domain. Furthermore, migrant patients experienced lower satisfaction across several domains. These intricate findings suggest that hospitals should tailor their services and evaluation metrics to specific patient demographics, and that the government should adopt policies that reduce disparities in healthcare access and affordability to ultimately improve the satisfaction of vulnerable groups.

## 1. Introduction

In the past twenty years, China has significantly advanced health insurance coverage for its rapidly expanding and urbanizing population, covering over 95% of its population of 1.37 billion by 2011 [1]. Despite these advancements in healthcare coverage, patient–provider relationships are becoming increasingly strained and violent, underscoring deeper patient dissatisfaction and mistrust of the healthcare system [2,3,4]. In addition, the demand for quality health care is much higher than its supply, with physicians in tertiary hospitals averaging only around 2.4 min for each patient visit [5].

These healthcare tensions also exist within a landscape of stark income inequalities and unequal healthcare access. China’s income inequality is high, both in comparison to China’s past and to other countries at similar stages of economic development [6]. These income inequalities are especially concentrated along urban–rural and coastal–island divisions, with urban/coastal residents having significantly higher per capita income than their rural/inland counterparts [7,8,9]. Additionally, Chinese health insurance is aligned to a resident’s urban–rural household registration, also known as hukou, and is categorized into one of three major categories: Urban Employee Basic Medical Insurance (UEBMI) for urban employees, Urban Resident Basic Medical Insurance (URBMI) for urban residents who are unemployed or work in the informal sector, and the New Rural Cooperative Medical Insurance (NRCMI) for rural residents [10]. Indeed, this household registration-aligned health insurance system potentiates differential healthcare access [7,9,11,12,13].

To reduce these health disparities and strengthen patient–provider relationships, the government established a national healthcare improvement initiative in 2015, aiming to give patients a tangible sense of reduced process and improved quality of care. The Chinese government then commissioned the Peking Union Medical College School of Public Health to perform an annual third-party academic evaluation of its healthcare improvement initiative, known as the China National Patient Survey [14]. Though this interest in evaluating patient satisfaction is part of an increasing global effort to collect and improve upon patient satisfaction metrics, the theory behind patient satisfaction is multidimensional and not yet well understood, particularly within international contexts [15].

Current models of patient satisfaction theory present a patient’s healthcare satisfaction as the combination of patient intrinsic factors, expectations, and subjectively perceived outcomes—all three rooted in the unique cultural contexts through which they exist [16]. Of these three, patient intrinsic factors, such as patient sociodemographics, are generally only reported as part of the descriptive analysis and not centered in studies of patient satisfaction. Furthermore, while certain studies acknowledge the influence sociodemographic characteristics such as age, gender, education, and health status can have on a patient’s satisfaction [17,18], these studies rarely delve deeper into how the sociodemographic characteristics shape individual sub-components of patient satisfaction. Lastly, sociodemographic trends are highly variable across regions and hospitals, and often report conflicting results. These discrepancies are likely due to the complex differences in macro-level processes in individual healthcare delivery systems, as well as the limited scope through which these studies are performed [5].

Broad-scale patient satisfaction studies at the national healthcare system level are scarce within China. Current Chinese studies reported in international research arenas have been either theoretical studies or survey analyses limited to one hospital or one city, and are therefore not particularly generalizable to other hospitals across China [19]. Also, because comparably fewer studies on Chinese patient satisfaction are reported in English, the international research community has limited access to and understanding of Chinese patient satisfaction. Our study builds upon the original China National Patient Survey results by focusing on how outpatients’ intrinsic sociodemographic characteristics correlate with key domains of patient satisfaction—process management, diagnosis and treatment, hospital environment, affordability, and overall satisfaction. We have selected variables that are known to influence an outpatient’s expectations, trust, adherence to medical recommendations, perceived symptom resolution, and ultimate healthcare satisfaction [19,20,21,22,23]. The objective of the study is to characterize how national Chinese outpatient socio-demographics impact their satisfaction across multiple healthcare domains.

## 2. Materials and Methods

### 2.1. Data

The study analyzes data from the 2017 China National Patient Survey, conducted by the Peking Union Medical College (PUMC) School of Public Health. The analyzed survey data were collected from December 2017–January 2018 in 136 tertiary hospitals across 31 provinces. Within each province, one provincial general hospital, one provincial traditional Chinese medicine hospital, and one maternal and child health hospital were selected. The remaining 43 hospitals included were National Health and Family Planning Commission (NHFPC)-affiliated hospitals (including 28 general hospitals and 15 specialist hospitals). China, which lacks a strong primary care system and gate-keeping mechanism, utilizes hospitals as a large provider for its outpatient service. In 2017, 42.1% of overall outpatient visits were provided by hospitals [24]. These sample hospitals accounted for 6.4% of outpatient visits of all the hospitals, and 2.7% of total outpatient visits nationwide.

The respondents were randomly selected in the pharmacy area of the hospital, where they usually complete the outpatient consultation and payment process and await receiving drugs. Patients of emergency departments and Very Important Person(VIP) Clinics (a special outpatient department with shorter waiting time, senior physicians, and higher charges) were excluded because of their special characteristics. Interviews were conducted face-to-face with pre-trained medical students via mobile devices. Only fully completed questionnaires could be successfully submitted online. We supposed that 85% of outpatients were satisfied with the hospital service, and set the significance level at 0.05 to calculate the minimum sample size, which was 196 for each hospital. Thus, the sample size was set as 200 per sample hospital. A total of 28,822 outpatient questionnaires were completed in the 2017 China National Patient Survey, and the response rate was 73.19%. Considering that the average number of daily outpatient visits per sampled hospital is 4487 (calculated from operational data reports by the sample hospitals), a sample size of 200 represents about 5% of daily visits. Before data analysis, we cleaned the data for erroneous entries with illogical relationships between demographic and social characteristics and gained a sample of 28,760 effective responses (99.78% effective rate).

### 2.2. Meausures

The outpatient sociodemographic variables of interest were age, gender, education, income, occupation, insurance, household registration, and patient source. Age was categorized as 18–35, 35–65, or >65 (no outpatient data for patients <18 was used in this study). Gender was defined as either male or female. Income categories included 0–60,000 RMB, 60–120,000 RMB, or >120,000 RMB. The level of education was reported in categories of either middle school/elementary school, high school, or undergraduate and above. The occupation categories were consolidated to unemployed/retired, public sector, and non-public sector employment. Outpatient insurance categories included Government Insurance Scheme (GIS), Urban Employees Basic Medical Insurance (UEBMI), Urban and Rural Residents Basic Medical Insurance (RBMI), New Rural Cooperative Medical Insurance (NRCMI), commercial insurance, medical aid, or uninsured. Household registration was classified as urban or rural. Patient source referred to local (patient residing in the hospital’s city with local household registration), local migrant (patient residing in the hospital’s city without local household registration), or non-local patient (patient residing outside of the hospital’s city).

Outpatient satisfaction was measured on a Likert scale of either 1, 2, 3, 4, or 5, corresponding respectively with ‘strongly disagree’, ‘disagree’, ‘neither agree nor disagree’, ‘agree’, and ‘strongly agree’. There were 23 items in the questionnaire regarding patient satisfaction, which were grouped into four key domains: process management, diagnosis and treatment, hospital environment, and affordability. The Kaiser–Meyer–Olkin value was 0.945, and the χ^2^ value of Bartlett’s test of sphericity was 51,090.083 (*p* < 0.001), indicating that the data were adequate for factor analysis. Exploratory factor analysis was conducted to evaluate the construct validity of the questionnaire. There were four factors with eigenvalue >1, and the cumulative variance percentage was 59.94%. The factor loading of each item was larger than 0.50, which indicated that the construct validity of the questionnaire was acceptable. For each of the four domains, we calculated the mean score of the items as a domain level satisfaction score, and tested for internal consistency using Cronbach’s α coefficient. As shown in Table 1, the mean satisfaction score ranged from 3.34 to 4.34, with Cronbach’s α all at acceptable levels. In addition, we examined the overall patient satisfaction valuation.

### 2.3. Statistical Analysis

Descriptive analyses were conducted, and mean domain satisfaction score along with sociodemographic characteristics were reported. In addition, an ANOVA test was performed to determine whether significant differences existed within each sociodemographic variable.

Given the nationally comprehensive nature of the survey data and natural structure of patient information divided into individual hospital levels, we explored multilevel logistic regression as a model for more robustly understanding how outpatient sociodemographic factors influence their satisfaction [25]. Outpatient satisfaction scores were reorganized into binary outcomes, using top-2-box scoring to redefine ‘strongly agree’ and ‘agree’ as a single positive answer (1 = yes) and ‘neither agree nor disagree,’ ‘disagree’, and ‘strongly disagree’ as a single negative answer (0 = no) [26]. Outpatient survey data were analyzed at both the patient and hospital levels. Using the MLwiN 2.02 software package (statistical software for multilevel models), we found that inter-cluster correlation coefficients (ICCs) at the hospital level ranged from 12.4% to 28.6% across all domains and dimensions, indicating that a significant portion of the variance in outpatient satisfaction is clustered at the hospital level [27,28]. Because the focus of our study is on the intrinsic patient, rather than hospital-level factors, we present the results of a modified multilevel logistic regression with fixed hospital effects to focus on outpatient satisfaction within each hospital. In the case of national data clustered at the hospital level, a multilevel logistic regression corrects for the standard error underestimation offered by traditional one-level regression analyses [27].

### 2.4. Ethics Statement

The study protocol was approved by the Ethics Committee of Peking Union Medical College (SPH201712CHII206). The survey of patient satisfaction was anonymous. Informed consent of each respondent was obtained before the survey.

## 3. Results

According to descriptive analysis, most patients surveyed were female (64.57%), local patients (62.82%) with urban household registration (72.43%), aged 18–35 (50.28%), having an income of 0–60,000 RMB (54.54%), an undergraduate degree or above (58.94%), and who were non-public sector employees (54.82%). The most commonly reported insurances were either the Urban Employees Basic Medical Insurance (31.31%) or the Urban and Rural Residents Basic Medical Insurance (30.65%) (Table 2).

Table 3 shows the mean satisfaction scores for each domain classified by sociodemographic groups. The highest reported mean score came from overall satisfaction (4.37) while the highest mean domain score was in diagnosis and treatment (4.34). Conversely, the lowest mean domain score was in process management (3.34). On the basis of the ANOVA analysis, all group means were statistically different (*p* < 0.05) from one another except income with respect to hospital environment, and education with respect to patient–physician relationship.

Under Table 4, the multilevel logistic regression analysis, patients who were local migrants, of highest income, most educated, and who had medical aid insurance reported the lowest levels of overall satisfaction within their respective sociodemographic categories. Rural outpatients showed lower (though not statistically significant) levels of overall satisfaction (Odds Ratio (OR) = 0.90, *p* = 0.068). Patients who were non-local or local migrants reported less satisfaction than local residents across all domains and in overall satisfaction, except for hospital environment. Local migrant patients reported the lowest overall satisfaction (OR = 0.82, *p* = 0.001), process management (OR = 0.89, *p* = 0.010), and diagnosis and treatment behaviors (OR = 0.83, *p* < 0.0001). Non-local patients reported the lowest satisfaction in affordability (OR = 0.88, *p* = 0.003). With regard to age, while there was no difference in age on an outpatient’s overall satisfaction, older patients over the age of 65 reported significantly greater satisfaction compared with patients aged 18–35 and 35–65 in diagnosis and treatment (OR = 1.24, *p* = 0.006), and the lowest satisfaction in process management (OR = 0.73, *p* = 0.002). Patients aged 35–65 reported the lowest satisfaction in affordability (OR = 0.91, *p* = 0.012), with no other significant differences in overall or individual domain scores. In terms of education, patients with an undergraduate degree and above reported the highest levels of satisfaction compared with other educational levels for process management (OR = 1.24, *p* = 0.0003) and affordability (OR = 1.11, *p* = 0.047). On the other hand, highly educated individuals reported the lowest satisfaction for diagnosis and treatment (OR = 0.85, *p* = 0.002), hospital environment (OR = 0.85, *p* = 0.001), as well as overall satisfaction (OR = 0.85, *p* = 0.031). Income correlated with outpatient satisfaction much in the same way as educational status. Compared to the poorest patients earning 0–60,000 RMB annually, the richest patients (annual income >120,000 RMB) reported the highest satisfaction in affordability (OR = 1.44, *p* < 0.0001) and second highest satisfaction in process management (OR = 1.15, *p* = 0.001). On the other hand, richer patients also reported lowest scores in hospital environment (OR = 0.78, *p* < 0.0001) and overall satisfaction (OR = 0.82, *p* = 0.0003). With regard to occupation, public sector employees reported the highest process management scores (OR = 1.28, *p* < 0.0001) compared with non-public sector employees (OR = 1.12, OR = 0.033) and reference unemployed/retired outpatients. Public sector employees additionally reported the highest scores in hospital environment (OR = 1.17, *p* = 0.001). In terms of insurance, when compared to the uninsured/retired outpatient reference group, highest rates of overall satisfaction for insurance were reported by patients with UEBMI (OR = 1.25, *p* = 0.005) and NRCMI (OR = 1.27, *p* = 0.006). Additionally, all outpatients, except those with commercial insurance and medical aid, reported higher satisfaction in terms of affordability, with GIS patients reporting the highest (OR = 1.33, *p* = 0.0002). Lastly, both GIS (OR = 1.19, *p* = 0.007) and UEBMI (OR = 1.13, *p* = 0.016) patients reported higher satisfaction scores with respect to hospital environment. Conversely, the lowest rate of overall satisfaction was from patients receiving highly limited services under medical aid (OR = 0.52, *p* = 0.032). Individuals with medical aid further reported lowest satisfaction scores in diagnosis and treatment (OR = 0.52, *p* = 0.011).

## 4. Discussion

### 4.1. Gender

Gender was the only variable which showed no significant association with any of the individual domains or with overall satisfaction. Indeed, previous studies of Chinese patient satisfaction confirm that the effects of gender on patient satisfaction are highly variable, with some papers showing female outpatients to be more satisfied [28,29,30,31], others showing higher satisfaction in males [5,19,32], and others even reaffirming no difference between male and female patients [14,33].

### 4.2. Age

Previous studies across China generally find that older patients tend to report higher satisfaction [19,30,34]. The lower reported satisfaction scores in process management may be due in part to questions in this domain, which ask about the satisfaction with use of technology in registration, bill payment, test result access, and pharmacy education. Older Chinese users report increased difficulty in accessing Internet-based resources and technologies; technology accessibility for this demographic must be considered in future hospital reforms aimed at improving satisfaction among older patients [35].

### 4.3. Education

Our results are concordant with previous studies that report lower satisfaction among more educated Chinese patients [5,19,32,34,36]. Our study highlights that dissatisfaction among educated patients may be more related with communication with the provider and hospital environment than process management issues such as waiting times and technology use.

### 4.4. Income

The trend in higher income being associated with higher affordability satisfaction and lower overall satisfaction levels is consistent with outpatient studies at various regional levels across the country [5,19,32,34]. Much like in the case of educational status, while lower-income outpatients might have lower expectations regarding their medical care, high-income outpatients may conversely have higher expectations for quality—both in their medical care and in the physical environment through which they receive their care [19].

### 4.5. Household Registration

Despite the aforementioned high income equalities between urban and rural populations, patients with rural household registration did not report significantly different satisfaction from urban patients across all domains. Furthermore, Chinese outpatient satisfaction scores have been shown to be higher in rural patients at the regional level [30,34]. However, this higher satisfaction among rural patients may be transient, as rural patients have only more recently enjoyed healthcare reforms that have expanded their access to medical infrastructure [30]. As rural outpatient expectations rise to meet those of their urban counterparts, their satisfaction may soon reflect still-existing socioeconomic disparities. The rural outpatients’ already existing lower overall satisfaction merits further investigation into what sorts of barriers exist for rural patients seeking care in tertiary hospitals.

### 4.6. Patient Source 

Lower satisfaction is expected from patients without the appropriate household registration because of the importability of social medical insurance. Non-local patients and local migrant patients utilize fewer healthcare resources and pay significantly more out-of-pocket for the care they receive compared with local residents with household registration [37]. As China’s rapidly developing economy continues to drive rural migrants into urban areas where they might lack access to local household registration, future health and social policies should expand upon immigrant social welfare coverage to address these inequities.

### 4.7. Occupation

Public sector employee’s higher satisfaction with their healthcare experiences in public hospitals is in line with their reported higher job satisfaction in general, likely due to greater job stability with less effort required compared with private sector employees [38,39]. Interestingly, there were no significant differences among occupations with regards to perceived affordability, diagnosis and treatment and overall satisfaction.

### 4.8. Insurance

Generally, patients who receive Urban Employee Basic Medical Insurance (UEBMI), such as most formal sector employees, enjoy the most comprehensive coverage of the available types of insurance. The Residents Basic Medical Insurance (RBMI) and New Rural Cooperative Medical Insurance (NRCMI) are voluntary programs that are highly subsidized by the government, but have more shallow coverage, especially for chronic or other outpatient expenditures. In addition to these three major insurances, surveyed patients could also report Government Insurance Scheme (GIS—health insurance at the state’s expense), private commercial insurance, medical aid, or uninsured. Thus, GIS and UEBMI patients reporting higher satisfaction scores with respect to hospital environment may be a reflection of the broader coverage and hospital access that these types of insurance afford their users.

As mentioned above, the lowest rate of overall satisfaction was from patients receiving highly limited services under medical aid. However, because the sample size for patients on medical aid was so small (*n* = 77), these trends must be cautiously interpreted. Otherwise, insurance type does not seem to play a significant role in influencing outpatient satisfaction in the process management domain. Thus, future national insurance reform should continue to focus more on expanding coverage and equalizing benefits as ways to improve outpatient satisfaction for all of its healthcare users.

The previous non-multilevel analysis of the 2015 China National Patient Survey outpatient sociodemographic variables only focused on overall patient satisfaction and did not report effects of these variables on individual satisfaction domains, which unfortunately limits the time-series comparisons that can be done. Within the 2015 analyses of overall outpatient satisfaction, only commercial insurance coverage was statistically significant (OR = 1.73, *p* = 0.03) [14]. However, our 2017 multilevel logistical regression analysis of outpatient sociodemographic variables revealed more significance in patients’ household registration, age, income, education, and insurance correlating with overall satisfaction. Furthermore, our study demonstrated the value of breaking down satisfaction into individual domains to elucidate finer-detailed relationships between a patient’s intrinsic factors and their overall satisfaction.

### 4.9. Limitations

The first limitation of our study is that we did not collect information about patient disease status or health outcomes, which are known influencers of patient satisfaction. Secondly, while we dichotomized the data using an established top-box approach for statistical analysis, this form of data aggregation can potentially mask subtler trends in China’s outpatient satisfaction. Thirdly, though this is the first major national study of sociodemographic influencers of outpatient satisfaction, only three hospitals from each province were included as samples (except Beijing, Shanghai, Guangdong, and Sichuan). Furthermore, these were all top teaching hospitals located in urban areas, and the distribution of patients surveyed was not even, for example, with 72.4% urban to 27.6% rural. This is not representative of the 2017 national ratio of 57.4% urban to 42.6% rural [40]. Fourthly, in the on-site survey, 26.81% of respondents refused to take the survey. Given that the characteristics of these patients are unknown, the findings in this study may only reflect the patients willing to participate and thus cannot be generalized to the whole population of patients. Lastly, this study only reveals the effect of sociodemographic factors on satisfaction and does not interrogate its underlying mechanism. The patients’ expectation, special needs (for example, elder patients with technology), and accessibility issues might all impact satisfaction, which requires further research.

## 5. Conclusion

China’s increasing attention to patient satisfaction evaluation is part of an international trend in developing patient-centered care. Our multilevel logistic analysis revealed that the patients who reported lowest levels of overall satisfaction were local migrants, of highest income and education, or who had medical aid insurance. Furthermore, our study revealed several more intricate relationships between a patient’s intrinsic sociodemographic variables and the individual domains of satisfaction. First, older outpatients were less satisfied with process management, possibly due to less familiarity with advancing information technologies. Next, since high-income and high-education outpatients reported lower satisfaction scores, more research is needed to identify their demand and expectations to provide diversified service. Lastly, patients without a local household registration experienced lower satisfaction across several domains, indicating access/affordability issues with this migrant population. These robust understandings can be used to more optimally monitor and assess specific outpatient satisfaction domains for disparities between population groups.

We also suggest that the Chinese government adopt policies to reduce disparities in healthcare access and affordability and ultimately improve satisfaction of vulnerable groups. For example, hospital-level cultural humility and implicit bias trainings have been employed in the United States to address provider biases in their process of diagnosis and treatment, and may be fruitful with regards to China’s rural and non-local patients [41]. Furthermore, patient navigation services have been effective in reducing health disparities, particularly among some of the United States’ most complex patient populations [42], which could also be implemented in China. In addition, focus groups and community advisory boards with older patients and patients of highest income and education may be necessary to understand their unique expectations. Lastly, social welfare programs such as medical aid insurance should be evaluated to understand why those patients reported lower satisfaction scores. Indeed, future hospital-level interventions should be careful to include a wide variety of sociodemographic variables. Our comprehensive study suggests that in the transition towards patient-centered healthcare, tertiary hospitals should apply greater consideration to how unique sociodemographic factors impact outpatients’ satisfaction, and make improvements accordingly.

## Figures and Tables

**Table 1 ijerph-16-03518-t001:** Content of the four domains and overall satisfaction, internal consistency, and mean domain score.

Domain	Items	Content	Cronbach’s α	Mean Score
Process management	10	Waiting time for registration/consultation/ medical tests/drug dispensing/paying bills, consultation length, convenience in making appointments, convenience in paying bills, convenience in printing reports, promotion of reasonable use of drugs	0.782	3.34
Diagnosis and treatment	8	Inquiry of symptoms with patience, explaining test results in detail, explaining treatments and medications with patience, feeling respected by physicians and nurses, privacy protection, attitude of staff at triage desk, doctor’s professional skills, timely guidance from the staff when needed	0.902	4.34
Hospital environment	4	Convenience of elevator, cleanliness of bathrooms, enough chairs in the waiting zones, drinking water supplies in the waiting zones	0.753	3.95
Affordability	1	The cost of this visit is within my ability to pay	-	4.18
Overall satisfaction	1	Overall I am satisfied with this visit	-	4.37

**Table 2 ijerph-16-03518-t002:** Descriptive summary of key sociodemographic outpatient variables.

Variables	*N* (%)
**Gender**	
Male	10,189 (35.43%)
Female	18,571 (64.57%)
**Household Registration**	
Urban	20,832 (72.43%)
Rural	7928 (27.57%)
**Patient Source**	
Local	18,067 (62.82%)
Local migrant	4458 (15.50%)
Non-local	6235 (21.68%)
**Age**	
18–35	14,461 (50.28%)
35–65	12,005 (41.74%)
> 65	1826 (6.35%)
**Income**	
0–60,000 RMB	15,686 (54.54%)
60–120,000 RMB	6954 (24.18%)
> 120,000 RMB	6120 (21.28%)
**Highest Level of Education**	
Undergraduate and above	16,951 (58.94%)
High school	7391 (25.70%)
Middle school/elementary school and below	4417 (15.36%)
**Occupation**	
Unemployed/retired	4613 (16.04%)
Public sector	8189 (28.47%)
Non-public sector	15,766 (54.82%)
**Insurance**	
Uninsured	2714 (9.44%)
Government Insurance Scheme (GIS)	3608 (12.55%)
Urban Employees Basic Medical Insurance (UEBMI)	9006 (31.31%)
Urban and Rural Residents Basic Medical Insurance (RBMI)^1^	8815 (30.65%)
New Rural Cooperative Medical Insurance (NRCMI)	3774 (13.12%)
Commercial insurance	538 (1.87%)
Medical aid	77 (0.27%)
**Total**	**28,760**

^1^ Includes Urban Resident Basic Medical Insurance (URBMI) and a merged plan of URBMI and NRCMI in some regions referred to as Urban and Rural Residents Basic Medical Insurance.

**Table 3 ijerph-16-03518-t003:** Mean domain scores (ANOVA *p*-values reported, *p*-value < 0.05 considered statistically significant).

	Process Management		Diagnosis and Treatment		Hospital Environment		Affordability		Overall Satisfaction	
	Mean Score	*p*-value	Mean Score	*p*-value	Mean Score	*p*-value	Mean Score	*p*-value	Mean Score	*p*-value
Gender										
Male	3.29	<0.001	4.34	<0.001	3.94	0.027	4.17	0.01	4.36	<0.001
Female	3.4		4.36		3.97		4.2		4.39	
Patient Source										
Local	3.38	<0.001	4.38	<0.001	3.97	<0.001	4.25	<0.001	4.41	<0.001
Local migrant	3.36		4.26		3.89		4.11		4.29	
Non-local	3.31		4.34		3.96		4.1		4.37	
Household Registration										
Urban	3.38	<0.001	4.37	<0.001	3.98	<0.001	4.24	<0.001	4.4	<0.001
Rural	3.3		4.29		3.91		4.07		4.33	
Age										
18–35	3.46	<0.001	4.34	0.006	3.94	<0.001	4.21	0.001	4.36	<0.001
35–65	3.29		4.36		3.97		4.17		4.39	
>65	3.04		4.38		4.04		4.14		4.45	
Income										
0–60,000 RMB	3.31	<0.001	4.34	<0.001	3.96	0.843	4.12	<0.001	4.37	0.002
60–120,000 RMB	3.4		4.37		3.96		4.24		4.4	
>120,000 RMB	3.45		4.36		3.94		4.33		4.39	
Highest Level of Education										
Undergraduate and above	3.44	<0.001	4.36	<0.001	3.97	<0.001	4.25	<0.001	4.39	<0.001
High school	3.33		4.34		3.96		4.16		4.36	
Middle school/elementary school	3.1		4.33		3.89		4.03		4.37	
Occupation										
Unemployed/retired	3.14	<0.001	4.35	<0.001	3.93	<0.001	4.12	<0.001	4.40	<0.001
Public sector	3.45		4.40		4.04		4.29		4.43	
Non-public sector	3.38		4.33		3.92		4.16		4.35	
Insurance										
Uninsured	3.33	<0.001	4.33	<0.001	3.87	<0.001	4.12	<0.001	4.36	<0.001
Government Insurance Scheme (GIS)	3.46		4.44		4.11		4.34		4.47	
Urban Employees Basic Medical Insurance (UEBMI)	3.38		4.35		3.99		4.23		4.4	
Urban and Rural Residents Basic Medical Insurance (RBMI)	3.36		4.34		3.92		4.19		4.36	
New Rural Cooperative Medical Insurance (NRCMI)	3.25		4.3		3.89		4.03		4.32	
Commercial insurance	3.45		4.33		3.91		4.22		4.35	
Medical aid	3.38		4.18		3.85		4.08		4.13	
**Total mean**	3.34		4.34		3.95		4.18		4.37	

**Table 4 ijerph-16-03518-t004:** Multilevel logistic regression analysis of domain scores with key socio-demographic outpatient variables. (*p*-value < 0.05 bolded and considered statistically significant)

	Process Management		Diagnosis and Treatment		Hospital Environment		Affordability		Overall Satisfaction	
	*p*-value	OR	*p*-value	OR	*p*-value	OR	*p*-value	OR	*p*-value	OR
**Gender**										
Male	Reference									
Female	0.094	1.06	0.793	1.01	0.660	0.99	0.460	0.98	0.758	1.01
**Household Registration**										
Urban	Reference									
Rural	0.091	0.93	0.134	0.94	0.885	1.01	0.209	0.94	0.068	0.90
**Patient Source**										
Local	Reference									
Local migrant	**0.010**	0.89	**<0.001**	0.83	0.769	0.99	0.007	0.88	**<0.001**	0.82
Non-local	0.174	0.94	**0.043**	0.92	0.596	0.98	0.003	0.88	0.779	0.98
**Age**										
18–35	Reference									
35–65	**<0.001**	0.83	0.394	1.03	0.958	1.00	0.012	0.91	0.476	0.97
>65	**<0.001**	0.73	**0.006**	1.24	0.052	1.14	0.290	0.92	0.244	1.13
**Income**										
0–60,000 RMB	Reference									
60–120,000 RMB	**<0.001**	1.15	0.614	1.02	**<0.001**	0.88	<0.001	1.18	0.365	1.05
>120,000 RMB	**<0.001**	1.15	0.623	0.98	**<0.001**	0.78	<0.001	1.44	**<0.001**	0.82
**Education**										
Middle school/elementary school	Reference									
High school	**0.026**	1.14	**0.016**	0.88	0.291	0.95	0.640	1.02	0.077	0.88
Undergraduate and above	**<0.001**	1.24	**0.002**	0.85	**0.001**	0.85	0.047	1.11	**0.031**	0.85
**Occupation**										
Unemployed/retired	Reference									
Public sector	**<0.001**	1.28	0.072	1.11	**0.001**	1.17	0.097	1.10	0.073	1.15
Non-public sector	**0.033**	1.12	0.966	1.00	0.108	1.07	0.503	1.03	0.880	1.01
**Insurance**										
Uninsured	Reference									
Government Insurance Scheme (GIS)	0.130	1.11	0.236	1.09	**0.007**	1.19	<0.001	1.33	0.098	1.18
Urban Employees Basic Medical Insurance (UEBMI)	0.268	1.07	0.421	1.05	**0.016**	1.13	<0.001	1.26	**0.005**	1.25
Urban and Rural Residents Basic Medical Insurance (RBMI)	0.747	1.02	0.218	1.07	0.179	1.07	<0.001	1.26	0.256	1.09
New Rural Cooperative Medical Insurance (NRCMI)	0.825	0.98	0.113	1.11	0.557	1.04	0.002	1.22	**0.006**	1.27
Commercial insurance	0.511	1.08	0.300	0.89	0.470	0.93	0.267	1.15	0.834	0.97
Medical aid	0.543	0.83	**0.011**	0.52	0.227	0.74	0.498	0.83	**0.032**	0.52

Bold characteristics indicate *p* value < 0.05.

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
