# Peer review of "Outpatient Satisfaction with Tertiary Hospitals in China: The Role of Sociodemographic Characteristics"

_ijerph, 2019, doi:10.3390/ijerph16193518_

Round 1
Reviewer 1 Report
A very important study in the Chinese context and it gives a good insight into the Chinese context. I have some specific points to be addressed before this can be published:
you mentioned 'illogical relationships' between variables and it would be helpful to give an example what is a 'hukou' - this will need to be explained you use the expression 'average' when you mean 'mean' in Table 3 you report p<0.000; it should be p<0.001 in the limitations mention that you had to dichotomize the satisfaction variable - you cannot guarantee that the responses would be the same if you actually requested a dichotomised responseAuthor Response
Point 1: You mentioned 'illogical relationships' between variables and it would be helpful to give an example what is a 'hukou' - this will need to be explained you use the expression 'average' when you mean 'mean' in Table 3 you report p<0.000; it should be p<0.001 in the limitations mention that you had to dichotomize the satisfaction variable - you cannot guarantee that the responses would be the same if you actually requested a dichotomised response
Response 1:
Thank you for the valuable feedback.
We’ve clarified that the data cleaning process was primarily to remove blatantly erroneous entries that did not make logical sense. Example includes male patients in OB/GYN department and was added in the text (p3, line 101-102). Hukou is the Chinese pronunciation for household registration and a unique word in China. We have a brief definition of hukou within the introduction (p1,line 40-41). To avoid difficulties for non-Chinese readers, we have changed ‘hukou’ to ‘household registration’ in the following text. We have changed the expression “average” to “mean” and also converted all p<0.000 to p<0.001 in Table 3. We have added a statement acknowledging the limitations of dichotomizing the data (p11,line 270-272).
Reviewer 2 Report
This is an interesting manuscript exploring socio-demographic and healthcare system variables associated with global and particular aspects of patient satisfaction.
The following issues should be addressed:
a) Methodology is unclear. It is stated that the survey was randomly administered. Still it is not specified 1) how the randomization was achieved 2) if and how many of the patients offered the survey refused to participate. This is obviously of importance since it is not clear if the findings are generalizable to the whole population of patients or they are just a reflection of patients willing to participate to the survey (and possibly biased in as much willing to participate). This should be also mentioned in the limitation section.
b) Authors should pay attention to awkward use of English language. For example "adherence to medical council " (page 2, line 71): here they probably mean "recommendations" o "prescriptions", "council" in this sentence is a mistake. Or "attitude of staff in service window" (Table 1). What do they mean ? Clinical staff at triage desk? Administrative staff at the counter?
c) some sentences just need to be much better explained because unclear. For example: (page 2, lines 93-94) "we cleaned the data for entries with illogical relationships etc". Authors need to make this sentence much more clear, giving examples if needed:
d) what VIP (page 2, line 87) means? Acronyms should be defined at their first use
Author Response
Point 1: Methodology is unclear. It is stated that the survey was randomly administered. Still it is not specified 1) how the randomization was achieved 2) if and how many of the patients offered the survey refused to participate. This is obviously of importance since it is not clear if the findings are generalizable to the whole population of patients or they are just a reflection of patients willing to participate to the survey (and possibly biased in as much willing to participate). This should be also mentioned in the limitation section.
Response 1: Thank you for the valuable comments.
1)In Chinese hospital outpatient department, patients usually wait to get drugs in the pharmacy area before leaving the hospital. These patients have completed the whole process of outpatient visit and are qualified for the survey. The investigators randomly select patients in the waiting zone of the pharmacy to conduct the survey, until the number of completed questionnaires has met the sample size of the hospital. We revised the description of the sampling method to be more clear (p2, line 84-85). 2) The response rate is 73.19% which we add in the data section (p3, line 91). And we also discuss it in the limitation section (p11, line 276-278).
Point 2: Authors should pay attention to awkward use of English language. For example "adherence to medical council " (page 2, line 71): here they probably mean "recommendations" o "prescriptions", "council" in this sentence is a mistake. Or "attitude of staff in service window" (Table 1). What do they mean ? Clinical staff at triage desk? Administrative staff at the counter?
Response 2: The terms used are adopted from previous studies whenever possible to remain consistent with the literature. However, we agree with the reviewer and have changed the specifically recommended terms to more clear (p2, line 70, Table 1).
Point 3: some sentences just need to be much better explained because unclear. For example: (page 2, lines 93-94) "we cleaned the data for entries with illogical relationships etc". Authors need to make this sentence much more clear, giving examples if needed:
Response 3: We’ve clarified that the data cleaning process was primarily to remove blatantly erroneous entries that did not make logical sense. Example includes male patients in OB/GYN department and was added in the text (p3, line 101-102).
Point 4: d) what VIP (page 2, line 87) means? Acronyms should be defined at their first use
Response 4: VIP here means “Very Important Person” which we originally had kept as an acronym given its more common usage. We have changed the term “VIP patients” to “VIP Clinic patients” and clarified its meaning on page 3, line 86.
Reviewer 3 Report
This paper is considered to be a low originality research and it is difficult to distinguish it from previous studies.Author Response
Point 1: This paper is considered to be a low originality research and it is difficult to distinguish it from previous studies.
Response 1: Thank you for your feedback. This is one of the few comprehensive analyses of national trends in China’s outpatient population, particularly with regards to how a patient’s many different sociodemographic factors can influence their satisfaction. The strengths of the paper lie in its data and in its ability to critique national trends, as the data was collected by the 2017 China National Patient Survey broadly from 136 tertiary hospitals across 31 provinces. We believe that this paper is particularly timely, both as China continues to launch major healthcare reforms with patient satisfaction in mind and given the popularity of international health disparities research.
We appreciate this reviewer’s feedback and hope that the collective edits made in this revision round will further distinguish our paper from previous studies.
Reviewer 4 Report
There are many ways to improve heath care services and the patients’ satisfaction is a relevant indicator. Nevertheless, these type of studies risk to be unnecessary without a policy determination and a systematic monitoring system even at different level of the organization and professionals to assess adverse working and environment conditions.
The paper could be publish without any changes, but if the authours could provide informations on a more comprehensive prevention activities in these health care structure it could be better!
Author Response
Point 1: There are many ways to improve heath care services and the patients’ satisfaction is a relevant indicator. Nevertheless, these type of studies risk to be unnecessary without a policy determination and a systematic monitoring system even at different level of the organization and professionals to assess adverse working and environment conditions.
The paper could be publish without any changes, but if the authours could provide informations on a more comprehensive prevention activities in these health care structure it could be better!
Response 1: Thank you for your valuable feedback. We have incorporated in our Conclusion section more concrete recommendations for health organizations given our national findings. We also are more explicit about how our recommendations can be used for future hospital-level policy determination and monitoring of patient satisfaction.